# Unsupervised Clustering of Patients Undergoing Thoracoscopic Ablation Identifies Relevant Phenotypes for Advanced Atrial Fibrillation

**DOI:** 10.3390/diagnostics15101269

**Published:** 2025-05-16

**Authors:** Ilse Meijer, Marc M. Terpstra, Oscar Camara, Henk A. Marquering, Nerea Arrarte Terreros, Joris R. de Groot

**Affiliations:** 1Biomedical Engineering and Physics, Amsterdam University Medical Centers, University of Amsterdam, 1105AZ Amsterdam, The Netherlands; h.a.marquering@amsterdamumc.nl; 2Radiology and Nuclear Medicine, Amsterdam University Medical Centers, University of Amsterdam, 1105AZ Amsterdam, The Netherlands; m.m.terpstra@amsterdamumc.nl; 3Department of Cardiology, Heart Center, Amsterdam University Medical Centers, University of Amsterdam, 1105AZ Amsterdam, The Netherlands; 4Physense, BCN MedTech, Department of Engineering, Universitat Pompeu Fabra, 08018 Barcelona, Spain; oscar.camara@upf.edu; 5Department of Digital Health & Biomedical Technologies, Vicomtech, 20009 Donostia-San Sebastián, Spain; narrarte@vicomtech.org

**Keywords:** unsupervised machine learning, principal component analysis, K-Means, thoracoscopic ablation, atrial fibrillation, phenotyping

## Abstract

**Background/Objectives**: The rate of recurrence after ablation for atrial fibrillation (AF) is considerable. Risk stratification for AF recurrence after ablation remains incompletely developed. Unsupervised clustering is a machine learning technique which might provide valuable insights in AF recurrence by identifying patient clusters using numerous clinical characteristics. We hypothesize that unsupervised clustering identifies patient clusters with different clinical phenotypes, including AF type and cardiovascular morbidities, and ablation outcomes. **Methods**: Baseline and procedural characteristics of 658 patients undergoing thoracoscopic ablation for advanced AF (persistent, with enlarged left atria, or with previous failed catheter ablation) between 2008 and 2021 were collected. Principal component analysis (PCA) was used as an unsupervised dimensionality reduction technique, followed by K-Means clustering for unsupervised data clustering. The silhouette score was used to determine the optimal number of clusters, resulting in the formation of three clusters. CHA_2_DS_2_-VASc score and AF recurrence were not included in the clustering, but were compared between clusters. Moreover, we compared the patients with and without previously established risk factors for AF recurrence for each cluster. **Results**: Unsupervised clustering resulted in three distinct clusters. Cluster I had a significantly lower rate of AF recurrence than Cluster II, which contained significantly more persistent AF patients than the other clusters. The CHA_2_DS_2_-VASc score in Cluster III was significantly higher than in the other clusters. In all clusters, but particularly in Cluster III, the recurrence risk was higher for persistent AF patients and female patients. In Cluster II, the recurrence risk was not influenced by an increased left atrial volume index, unlike other clusters. **Conclusions**: Using unsupervised clustering of clinical and procedural data, we identified three distinct advanced AF patient clusters with differences in AF type, CHA_2_DS_2_-VASc score, and AF recurrence. We found that established risk factors like BMI, AF type, and LAVI vary in importance across clusters.

## 1. Introduction

Atrial fibrillation (AF) is the most common cardiac arrhythmia, with a prevalence of 10–17% in people over 80 years old [1]. AF is characterized by disorganized electrical activity in both the atria, resulting in ineffective atrial contraction and an irregular, and often rapid, heart rate [2]. In symptomatic AF patients, there is an indication to restore sinus rhythm, and rhythm control strategies are applied. Rhythm control strategies comprise antiarrhythmic medication, cardioversion, or catheter/thoracoscopic ablation, depending on the patient’s characteristics. Catheter ablation is being performed increasingly for both patients with paroxysmal AF, i.e., AF episodes lasting less than seven days, regardless of previous antiarrhythmic drugs failure, and persistent AF, i.e., AF episodes lasting more than seven days or not terminating spontaneously [3]. Thoracoscopic surgical ablation is an alternative approach for patients with advanced AF, that is, patients with enlarged left atria, previously failed catheter ablation, or persistent AF [2,4,5].

Thoracoscopic ablation strategies consist of a bilateral, minimally invasive, video-assisted procedure using a bipolar radiofrequency ablation pen and a transpolar clamp for pulmonary vein isolation. These strategies have been described in detail before, with and without hybrid electrophysiological confirmation of ablation lesions [5,6]. Many studies have reported a higher freedom of AF after thoracoscopic ablation versus catheter ablation, with success rates ranging from 66 to 83% vs. 37 to 53% after one year of follow up, usually at the cost of more periprocedural complications [4,7,8]. There are also, however, randomized studies that show a lower success rate of thoracoscopic compared to catheter ablation in patients with longstanding persistent AF [9]. Thus, thoracoscopic ablation is clearly not suitable for all AF patients. It is important to avoid exposing those patients to complications that would not benefit from treatment and to avoid unnecessary healthcare costs. Therefore, identifying patients or groups of patients who will not benefit from thoracoscopic ablation is crucial.

Recently, multiple studies have applied unsupervised dimensionality reduction and clustering machine learning techniques for patient phenotyping and risk stratification in atrial fibrillation [10,11,12,13,14,15,16]. Specifically, unsupervised dimensionality reduction and clustering were used to identify clusters of patients based on a large number of characteristics [17]. Unsupervised clustering is the first step towards patient-specific treatment decisions, but does not aim to select patients for treatment directly. The main advantage of these unsupervised machine learning techniques is that many different types of data can be combined without the need for costly manual annotations and a priori assumptions, as such unsupervised machine learning allows us to detect unidentified patterns. These studies compared either outcome after catheter ablation or all-cause mortality, major bleeding, stroke, and myocardial infarction between clusters [10,11,12,13,14,15]. However, unsupervised machine learning has not yet been applied to clinical characteristics of patients with more advanced AF, such as those undergoing thoracoscopic ablation.

In this study, we applied unsupervised machine learning aiming to identify clusters of patients with advanced AF considering all clinical, procedural, and imaging data collected in routine clinical practice. We hypothesize that unsupervised machine learning will define the following:Clusters of patients with different outcomes of thoracoscopic ablation;Clusters of patients with differences beyond AF type and cardiovascular risk profiles (CHA_2_DS_2_-VASc score).

Moreover, we aim to assess the difference in established AF recurrence risk factors within clusters and to discern which risk factors are most relevant to each patient, thereby guiding treatment decisions. Patient stratification and understanding patient subgroups is the first step towards more personalized treatment selection.

## 2. Materials and Methods

### 2.1. Data Collection

Data for AF patients undergoing thoracoscopic ablation between February 2008 and December 2021 in the Amsterdam University Medical Center, a large tertiary center in the Netherlands, were collected. Patients that are selected for thoracoscopic ablation either have advanced AF or have expressed a preference for thoracoscopic ablation above other strategies. All patients provided informed consent and underwent thoracoscopic AF ablation with bilateral pulmonary vein isolation and left atrial appendage excision [5], as part of their inclusion in the AFACT trial [18], or in the MARK-AF registry study (NL50069.018.14). In patients with persistent AF, additional ablation lines were created [19]. Furthermore, patients who were included in the AFACT trial were randomized to either additional ganglion plexus ablation or not [18]. Our study population was followed up for two years by means of at least quarterly electrocardiograms and 24h Holter monitoring. All outcomes were adjudicated by an independent physician who was not part of the study group.

### 2.2. Data Preprocessing

The data were plotted and visually inspected for outliers or any data inconsistencies. Outliers or inconsistencies were removed after confirmation by an experienced cardiologist. Variables with over 30% missing values were excluded, since imputation methods have shown good performance, with up to 30% missing data [20,21]. Subsequently, patients with over 30% missing values were excluded following the same rationale. Variables were ignored when their Pearson correlation coefficient or their Cramer V coefficient with another variable was higher than 0.5 for numerical variables or categorical variables, respectively. Variables with the least missing data or clinician-preferred variables (e.g., Holter over electrocardiogram) were retained. If the point-biserial coefficient of the correlation between a numerical and dichotomous categorical variable was larger than 0.5, the categorical variable was removed. Variables that were combined with another variable to form a new variable were also discarded (for example, body mass index (BMI), which is the quotient of weight and length squared) [22]. Remaining missing values were imputed using MissForest, an iterative random forest-based imputation method [23], since it has been shown to achieve high-quality results up until 30% of missing data and to be superior in missing value imputation compared to other methods, such as predictive mean matching and k-nearest neighbors [21]. Moreover, it is able to handle both continuous and categorical data [23]. We standardized the included variables as follows: continuous variables were normalized between zero and one, and categorical variables were encoded using one-hot encoding. In the model input, we did not include the CHA_2_DS_2_-VASc score and the treatment outcome variables, namely 1-year and 2-year AF recurrence, since the CHA_2_DS_2_-VASc score is a point-based score on the risk of stroke in AF patients, and all factors that make up the score are included in the clustering.

### 2.3. Unsupervised Machine Learning

Our unsupervised machine learning approach consisted of two steps: dimensionality reduction using principal component analysis (PCA) and K-Means clustering. We selected PCA over other dimensionality reduction techniques because of its interpretability and stability [24,25]. We used K-Means clustering as our unsupervised clustering technique as it has been shown to optimize a similar objective to PCA, performs well in the low-dimensional space created by PCA, and provides interpretable clusters [26,27].

Dimensionality reduction reduces the number of variables by combining them into a smaller number of new variables. An intuitive example of this is the calculation of BMI as the quotient of weight and squared length of a patient. In unsupervised dimensionality reduction, it is not necessary to explicitly create the rules or formulas for the reduction in variables. To clearly display the new features, we chose to restrict the number of dimensions of the space to three. These low-dimensional datasets are more suitable as inputs for K-Means clustering than high-dimensional datasets. We considered that with the available number of patients, a higher number of dimensions would come at the cost of separation of the clusters. PCA is a linear dimensionality reduction technique that aims to transform the data into a new coordinate system, where the variance of data is maximized along the principal components. It identifies the directions (principal components) in which the data vary the most and projects the data onto these components. Each principal component is a linear combination of variables in the original dataset.

K-Means clustering divides a dataset into a predefined number of clusters [26]. The centroids of the clusters are initialized using sampling based on an empirical probability distribution of the points’ contribution to the overall inertia. Data points are iteratively assigned to the cluster with the nearest centroid, and the centroid of each cluster is updated based on the mean location of points in that cluster. The silhouette score was used to choose the best number of clusters [28].

### 2.4. Validation

To evaluate the quality of the dimensionality reduction, we calculated the percentage of explained variance of the three newly created principal components [29]. We assessed cluster separation using the silhouette score. The silhouette score is a metric that quantifies the separation of clusters by comparing the average distance within each cluster to the distance between different clusters. A silhouette score with a value of minus one indicates no separation, whereas a value of one indicates strong clustering. A silhouette score close to zero means that the groups are separated, but the boundaries of the groups are adjacent or overlap. We validate the clusters by creating 50 reduced models with random selections of 25% of the data omitted. We then re-clustered the remaining data and compared each model to the original model. Clusters were considered stable if there was high average agreement between the reduced and original models.

### 2.5. Statistical Analysis

The distribution of the patient characteristics for each cluster were summarized using descriptive statistics and compared using the appropriate statistical test. A radar plot was used to give a general description of the clusters and to make the clusters clinically interpretable. These plots were created using common characteristics in AF patients, such as age, BMI, and sex, as well as variables with a large effect size. For categorical variables, the effect size was considered large when Cohen’s w was greater than 0.5, and for numerical variables, when the rank biserial correlation was greater than 0.5 [30].

We assessed the prevalence of previously established clinical phenotypes within the clusters. Specifically, we investigated patients with paroxysmal or persistent AF and patients with low or high CHA_2_DS_2_-VASc scores. Moreover, we compared AF recurrence after thoracoscopic ablation between clusters. Finally, we calculated the relative risk in each cluster for well-known risk factors of AF recurrence, to assess whether such risk factors play a more prominent role in AF recurrence for different subpopulations of advanced AF, by determining the fraction of patients with recurrence per cluster in patients with and without a risk factor.

Numerical data were compared using the Kruskal–Wallis test, and reported as the median and interquartile range. Categorical data were compared using the χ^2^ test and reported as the number and percentage. If a significant difference between the clusters was found, a post hoc analysis was performed using the Mann–Whitney U test for numerical variables and the χ^2^ test for categorical variables. In both cases, multiple testing was corrected using the Bonferroni correction.

## 3. Results

### 3.1. Data Collection and Preprocessing

A total of 668 patients met our inclusion criteria. In total, 112 clinical variables were collected in routine clinical practice, including patient history, patient characteristics, procedure characteristics, treatment outcome, medication, Holter and electrocardiogram reports, parameters derived from imaging modalities (such as echocardiography, MRI, and CT), and laboratory measurements. Appendix A provides a comprehensive list of all variables used in the present study. After the exclusion of missing and correlated variables, 658 patients and 84 variables remained, with a median of 3% missing values (Appendix A, Appendix A). The final dataset consisted of 231 patients with paroxysmal AF, 409 with persistent AF, 18 with longstanding persistent AF, 489 males, and 169 females. In Appendix A, all the included baseline characteristics are summarized for the original dataset, the dataset in which outliers were removed, and the dataset that was imputed with MissForest. The differences between these datasets were not significant, except for the C-Reactive Protein value, which had a median of 1.8 in the imputed dataset and 1.4 in the original dataset (*p* < 0.05).

### 3.2. Cluster Creation and Validation

The three principal components found using PCA explain 13% of the variance in the complete dataset. A more detailed analysis of the principal components can be found in the Appendix A. We found the best separation for the number of clusters k = 3 consisting of 308, 188, and 162 patients. The silhouette score was 0.3, indicating that the clusters are separated but without ubiquitous borders for each cluster. The validation of the clusters showed an agreement of 96% (±2%) with the 50 validation models trained on a random selection of 75% of the data, indicating stability, as the clusters contained a large number of the same patients. Figure 1 shows the clusters plotted against the three principal components. Note that the clusters are well separated along the three principal components, meaning that the clusters consist of different patient phenotypes. In Appendix A, all clinical variables that were included in the clustering analysis are summarized for each of the three clusters. There was a significant difference between 61/84 variables for the three clusters, highlighting the clear distinction between the three clusters.

### 3.3. Clustering Analysis

In Figure 2, the differences between each cluster’s baseline characteristics are illustrated. Cluster I (*n* = 308) contained the lowest proportion of females (60/308, 19%), a lower BMI (27 kg/m^2^), and the lowest number of patients with hypertension (66/308, 21%). Patients in Cluster II (*n* = 188) had a higher heart rate (85 bpm), anterior–posterior (AP) length (46 mm), and proBNP value (802 ng/L). Cluster III (*n* = 162) contained almost exclusively patients with hypertension (145/162, 90%) and a higher proportion of females (55/162, 34%).

In Figure 3A,B, the distribution of AF type and CHA_2_DS_2_-VASc score for the three clusters is shown. Cluster II contained more persistent and longstanding persistent AF patients than the other clusters (86% vs. 47% and 63% for Cluster I and III, respectively, *p* < 0.0001). Patients in Cluster III had the highest CHA_2_DS_2_-VASc score: 42% of patients in Cluster III had a CHA_2_DS_2_-VASc > 2, while this amounted to only 5% in Cluster I and 18% in Cluster II (*p* < 0.0005).

Figure 4A,B shows the distribution of AF recurrence after thoracoscopic ablation outcomes for each cluster. We found that patients in Cluster II and Cluster III more often had recurrence of AF after one year than in Cluster I, although this difference was only significant for Cluster II (37% vs. 26%, *p* < 0.05 and 35% vs. 26%, *p* = 0.18, respectively). The AF recurrence rate after two years was 49% for Cluster II and 49% for Cluster III compared to 39% for Cluster I. However, this difference was not significant (*p* = 0.05). In Figure 4C,D, we show the difference in AF recurrence when separating patients on the single risk factor AF type. The AF recurrence rate after one year was 38% for persistent AF compared to 20% for paroxysmal AF patients (*p* < 0.0001) and 52% compared to 31% after two years (*p* < 0.0001). Thus, using all available data to create distinct patient groups with PCA and K-Means does not improve the AF recurrence risk estimation compared to single risk factors such as AF type.

In Table 1, we show the relative risk of previously established risk factors of AF recurrence in each cluster. We found that some risk factors are more indicative of AF recurrence in specific clusters. Specifically, persistent AF or being female is most indicative of AF recurrence in Cluster III. In Cluster II, the risk of AF recurrence is not increased for an increased left atrial volume index (LAVI), while it is in the other two clusters. An increased BMI only shows increased risk of AF recurrence in Cluster III, and a decreased risk of AF recurrence in Cluster I. 

## 4. Discussion

Automated unsupervised clustering identified three distinct advanced AF patient clusters varying in baseline and procedural characteristics. Cluster II consisted of more patients with persistent AF and Cluster III of more patients with a high CHA_2_DS_2_-VASc score compared to the other clusters. We found a higher rate of AF recurrence in Cluster II and III compared to Cluster I, although this difference was not bigger than when comparing based on AF type. We found an increased risk of AF recurrence for persistent AF patients and females in Cluster III compared to the other clusters. Furthermore, an increased LAVI did not increase the risk of AF recurrence in Cluster II, while it did in the other clusters. This highlights the value of analyzing patient groups with similar characteristics: to identify the relevant risk factors for individual patients.

Previous studies have applied unsupervised clustering for the phenotyping of AF patients, employing methods like K-prototype clustering [13,15], or hierarchical clustering [11,12], either directly or after a dimensionality reduction method [14]. Given the large number of variables in our study, we opted for dimensionality reduction to avoid the “curse of dimensionality” inherent in distance-based clustering algorithms, such as K-Means and K-Prototype [17].

Inohara et al. [14], Bisson et al. [12], and Vitolo et al. [16] identified a healthier cluster of younger male patients with few comorbidities, similar to the first cluster we found, despite our focus on an advanced AF population. This highlights the difficulty of identifying patients that might benefit specifically more from thoracoscopic ablation instead of catheter ablation based solely on the baseline characteristics. The second cluster consisted of patients with higher BMI, higher ProBNP values, and longer AP diameter. Bisson et al. [11] described a comparable cluster, albeit with lower values of ProBNP and a slightly higher CHA_2_DS_2_-VASc score. Our third cluster consisted mainly of older, female patients with hypertension, and aligns with results from Saito et al. [15], Deb et al. [10], and Vitolo et al. [16].

Notably, Bisson et al. [11] found the highest AF recurrence rate one year after catheter ablation in a cluster with predominantly persistent AF (95%) and the largest CHA_2_DS_2_-VASc score (3 ± 2). They found an intermediate 1-year AF recurrence rate in a cluster with 81% persistent AF patients and low CHA_2_DS_2_-VASc score (1 ± 1). In contrast, our second cluster, with the highest proportion of persistent AF (87%) and low CHA_2_DS_2_-VASc score, had the largest rate of AF recurrence one year after thoracoscopic ablation, while our third cluster with 65% persistent AF patients and the largest CHA_2_DS_2_-VASc score (2 ± 1) had a slightly lower 1-year AF recurrence rate. Differences in AF recurrence between our study and Bisson et al. [11] may stem from treatment differences, follow-up duration, and patient selection. Patients in their study underwent catheter ablation, which is a less invasive approach associated with higher recurrence rates and a greater proportion of first-time ablations, while patients in our study underwent thoracoscopic ablation. Only advanced AF patients are eligible for this procedure, resulting in less patient variability in our cohort. Additionally, our follow-up period was two years, compared to one year in the study of Bisson et al. [11].

To the best of our knowledge, this is the first study applying unsupervised clustering of advanced AF patients undergoing thoracoscopic ablation. After validation in external cohorts, such models can be used to map data from AF patients that are considered for thoracoscopic ablation onto the determined three-dimensional space and assign them to their corresponding cluster. If clusters are found with clear differences in treatment success, the patient’s probability of successful treatment can be estimated based on the assigned cluster, and treatment can be tailored to individual patients. For clusters with moderate treatment success variability, this should not be used as a diagnostic tool, but instead patient stratification and risk assessment should be considered a first step towards personalized treatment selection. In our study, classifying a patient by its cluster could highlight which risk factors of AF recurrence are most important for that patient. Further research could refine this approach by building cluster-specific treatment success prediction models, and enhancing the clustering model with a larger, more comprehensive baseline dataset, with more thorough follow-up to reduce missing data and achieve a more balanced distribution of male and female patients. Furthermore, the importance of risk factors per cluster should be tested in a validation cohort or a prospective study.

Although the dataset was large and consisted of over 600 patients, one of the largest studies on cluster analysis in AF ablation, it still has some limitations. First, a few patients and variables had to be discarded because of missing data, which is an intrinsic issue with follow up in clinical studies. Valuable information might be hidden in this missing data that could change the clusters. Second, the dataset was not balanced: it consisted of more males than females and more patients with persistent AF, which is the case in most AF ablation studies. This might lead the model to miss hidden patterns in the underrepresented classes (females and paroxysmal AF patients). Third, the number of patients included might not be sufficient to find all patterns in the data, especially considering the high number of patient characteristics included in the unsupervised clustering. Fourth, it remains unclear whether the significance of these clusters persists over the long term. We await results from ongoing long-term follow-up studies to validate these findings. Finally, the clustering results may not generalize to similar patients from different cohorts or different patient groups, particularly less severe AF forms, despite some similarities found in clusters of patients undergoing catheter ablation.

The PCA model used for dimensionality reduction could only explain 13% of the variance and could not clearly separate outcomes of thoracoscopic ablation. The low explained variance might be due to the complexity of our data. Data reduction is a trade-of between simplification and accuracy. Summarizing our data in three dimensions might be too limited for the complexity and extensiveness of the included data. Moreover, the PCA model can only capture linear relations between variables and will not capture more complex relations, which most likely are present in the data. However, non-linear dimensionality reduction methods, such as Kernel-PCA, are less easily interpretable, and the parameters of the model have to be tailored to each specific dataset [31]. The dataset consisted of both categorical and numerical variables. Since nominal categorical variables have no order, these variables had to be encoded using a one-hot-encoding to be able to use them in the PCA model. This has the disadvantage that the dimensionality of the data was further increased first before it was reduced.

The K-Means method has some drawbacks, as the number of clusters (K) needs to be selected manually using, for example, the silhouette score or the elbow method. However, these approaches can yield inconclusive results, since multiple values of K can have the same optimal silhouette score. Secondly, K-Means assumes that clusters have a round or spherical shape, which may oversimplify the true data distribution.

In this study, we identified three distinct clinical phenotypes of advanced AF patients undergoing thoracoscopic ablation with an unsupervised machine learning approach. There was a difference in AF recurrence following thoracoscopic ablation between these clusters. Furthermore, the clusters differed in AF type and CHA_2_DS_2_-VASc score, which underscores that our unsupervised method was able to learn complex patterns in data. Moreover, the importance of single risk factors of AF recurrence differed per cluster, highlighting the potential of these methods to improve understanding of AF recurrence.

## Figures and Tables

**Figure 1 diagnostics-15-01269-f001:**
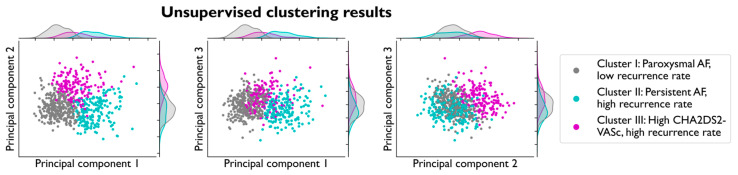
Patients represented as points in each pair of principal components. Each point is colored by the corresponding K-Means cluster of that patient. The distribution and separation of the clusters in each of the principal components can be visually analyzed using the distribution curves on the top and right side of the plot.

**Figure 2 diagnostics-15-01269-f002:**
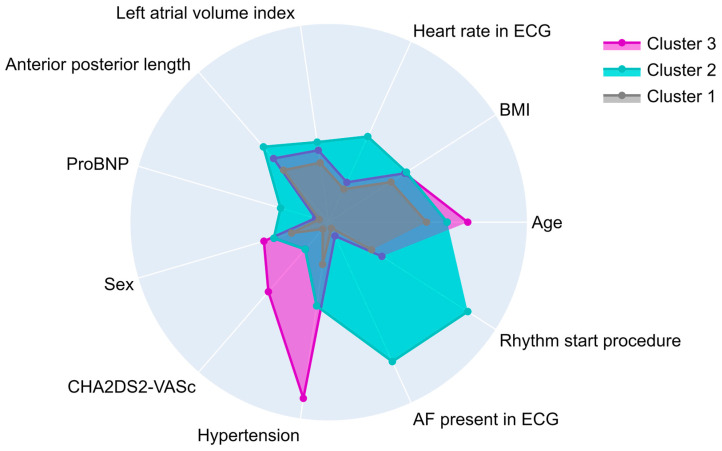
Comparison of the three clusters based on baseline and procedural characteristics. Cluster I: younger, more often male patients with lower BMI and less comorbidities. Cluster II: Patients with higher BMI, ProBNP values, and heart rate. Cluster III: Older, female patients with higher rate of hypertension. ProBNP—prohormone of brain natriuretic peptide.

**Figure 3 diagnostics-15-01269-f003:**
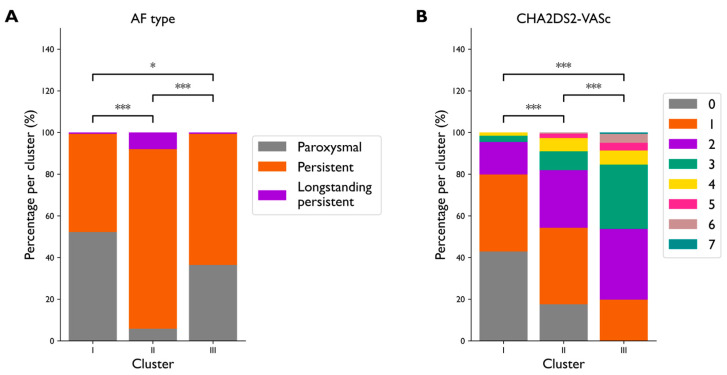
Distribution of the variables (**A**) AF type and (**B**) CHA_2_DS_2_-VASc score in the three clusters. Variables AF type and CHA_2_DS_2_-VASc score were not used to create the clusters. Note that Clusters I, II, and III contain 308, 188, and 162 patients, respectively. * *p* < 0.01, *** *p* < 0.0001.

**Figure 4 diagnostics-15-01269-f004:**
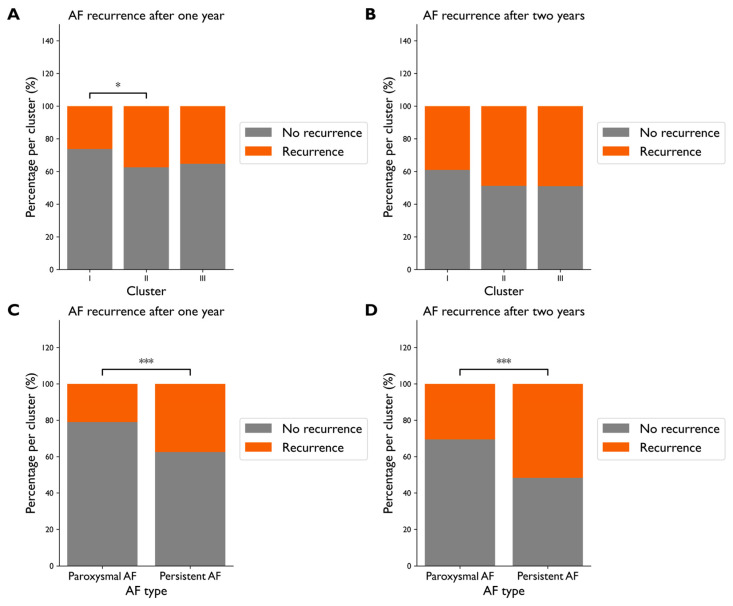
Distribution of AF recurrence: (**A**) 1-year and (**B**) 2-year AF recurrence in the three clusters, and (**C**) 1-year and (**D**) 2-year AF recurrence for paroxysmal compared to persistent AF. Note that Clusters I, II, and III contain 308, 188, and 162 patients, respectively. * *p* < 0.01, *** *p* < 0.0001.

**Table 1 diagnostics-15-01269-t001:** The relative risk of AF recurrence in each cluster for patients with and without well-known risk factors for AF recurrence, namely persistent AF, female sex, increased LAVI, and increased BMI.

Cluster	Risk of AF Recurrence		Relative Risk
	Paroxysmal AF	Persistent AF	Persistent AF/Paroxysmal AF
I	0.30 (45/151)	0.49 (67/136)	1.65
II	0.13 (1/8)	0.51 (80/158)	4.05
III	0.35 (19/54)	0.57 (53/93)	1.62
	Male	Female	Female/Male
I	0.37 (85/228)	0.46 (27/59)	1.22
II	0.44 (53/121)	0.62 (28/45)	1.42
III	0.39 (38/97)	0.68 (34/50)	1.74
	LAVI ≤ 40	LAVI > 40	(LAVI > 40)/(LAVI ≤ 40)
I	0.34 (60/174)	0.49 (50/102)	1.42
II	0.48 (31/64)	0.51 (49/97)	1.04
III	0.43 (25/58)	0.54 (45/84)	1.24
	BMI ≤ 27	BMI > 27	(BMI > 27)/(BMI ≤ 27)
I	0.43 (68/159)	0.35 (44/127)	0.81
II	0.54 (33/61)	0.46 (48/105)	0.85
III	0.41 (26/63)	0.55 (46/83)	1.34

## Data Availability

The data presented in this study are not publicly available due to privacy and ethical restrictions. The principal components and clustering model obtained in this paper are available upon request.

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
