# Peer review of "Unsupervised Clustering of Patients Undergoing Thoracoscopic Ablation Identifies Relevant Phenotypes for Advanced Atrial Fibrillation"

_diagnostics, 2025, doi:10.3390/diagnostics15101269_

Round 1
Reviewer 1 Report
Comments and Suggestions for Authors
Dear Authors,
I have read your manuscript titled "Unsupervised clustering of patients undergoing thoracoscopic ablation identifies relevant phenotypes for advanced atrial fibrillation" very carefully. The study addresses an important clinical question by applying unsupervised machine learning techniques to identify distinct patient clusters in the context of advanced atrial fibrillation (AF) treatment. The use of principal component analysis (PCA) and K-Means clustering on a comprehensive dataset is innovative and provides valuable insights into patient phenotypes and their association with AF recurrence.
Kindly read through my suggestions below, and consider implementing it to further enhance your manuscripts' quality and make it ready for audience of this journal.
- Consider adding more detailed explanation of the unsupervised clustering process in the abstract section. For example, mentioning the number of clusters identified earlier might help readers better understand the results section.
- The introduction section presents a comprehensive information on atrial fibrillation (AF), but Authors should explain briefly what thoracoscopic ablation entails for those unfamiliar with the procedure. This will make the manuscript more accessible to a broader audience.
- Your method section, under "Data Preprocessing", the criteria for excluding variables with over 30% missing values is clear, but perhaps elaborate on why this specific threshold was chosen. Was this based on previous studies or statistical based Knowledge?
- Regarding the use of MissForest for imputing missing data, briefly add a sentence or two about its advantages compared to other imputation methods, especially given that your dataset had up to 30% missing data due to patients missing values.
- When discussing PCA and k-means clustering, it might be useful to include a brief comparison with other dimensionality reduction techniques, like t-SNE or UMAP so as to justify why PCA was selected. though you did talked about Kernel-PCA later, but introducing alternatives earlier could strengthen your methodology section.
- In the discussion section, expand a little more on how the findings align with or differ from Bisson et al.'s work mentioned in the discussion. Consider highlighting key contrasts which may offer deeper insight into potential reasons behind observed discrepancies.
- Authors should mention possible future directions involving longitudinal studies following patients post-thoracoscopic ablation longer than two years. Such investigations could reveal whether initial clustering persists over extended periods.
Author Response
We thank the reviewer for their time and constructive feedback. We will answer the reviewer in a point-by-point fashion and have made changes to the manuscript accordingly.
Comment 1: Consider adding more detailed explanation of the unsupervised clustering process in the abstract section. For example, mentioning the number of clusters identified earlier might help readers better understand the results section.
Response 1: We have added the following in the abstract (line 24-29): Principal Component Analysis (PCA) was used as an unsupervised dimensionality reduction technique, followed by K-Means clustering for unsupervised data clustering. The silhouette-score was used to determine the optimal number of clusters, resulting in formation of three clusters.
Comment 2: The introduction section presents a comprehensive information on atrial fibrillation (AF), but Authors should explain briefly what thoracoscopic ablation entails for those unfamiliar with the procedure. This will make the manuscript more accessible to a broader audience.
Response 2: We have added the following in the introduction (line 57-59): Thoracoscopic ablation strategies consist of a bilateral, minimally invasive, video-assisted procedure using a bipolar radiofrequency ablation pen and a transpolar clamp for pulmonary vein isolation. These strategies have been described in detail before, with and without hybrid electrophysiological confirmation of ablation lesions [5,6].
Comment 3: Your method section, under "Data Preprocessing", the criteria for excluding variables with over 30% missing values is clear, but perhaps elaborate on why this specific threshold was chosen. Was this based on previous studies or statistical based Knowledge?
Response 3: We have added the following in the methods section (line 114-116): Variables with over 30% missing values were excluded, since imputation method have shown good performance up until 30% missing data [20,21]. Subsequently, patients with over 30% missing values were excluded following the same rationale.
Comment 4: Regarding the use of MissForest for imputing missing data, briefly add a sentence or two about its advantages compared to other imputation methods, especially given that your dataset had up to 30% missing data due to patients missing values.
Response 4: We have specified some methods that performed worse than MissForest and gave an additional reason for using MissForest (line 125-129): Remaining missing values were imputed using MissForest, an iterative random forest based imputation method [23], which since it has been shown to achieve high quality results up until 30% of missing data and to be superior in missing value imputation compared to other methods, such as predictive mean matching and k-nearest neighbors [21]. Moreover, it is able to handle both continuous and categorical data [23].
Comment 5: When discussing PCA and k-means clustering, it might be useful to include a brief comparison with other dimensionality reduction techniques, like t-SNE or UMAP so as to justify why PCA was selected. though you did talked about Kernel-PCA later, but introducing alternatives earlier could strengthen your methodology section.
Response 5: Before selecting PCA, we internally evaluated alternative dimensionality reduction methods, including t-SNE, UMAP, kernel PCA, and multiple kernel learning (MKL). These methods resulted in comparable cluster separation but did not improve outcome stratification. We ultimately selected PCA for its greater interpretability and robustness. To maintain focus on the PCA-derived clusters, we chose not to include an in-depth comparison in the main text. However, if the reviewer considers it valuable, we are happy to include these results in an appendix. We have added a general justification of our choice for PCA and K-Means in the methods section (line 137-141): We selected PCA over other dimensionality reduction techniques, because of its interpretability and stability [24]. We used K-Means clustering as our unsupervised clustering technique, as it has been shown to optimize a similar objective to PCA, performs well in the low-dimensional space created by PCA, and provides interpretable clusters [25,26].
Comment 6: In the discussion section, expand a little more on how the findings align with or differ from Bisson et al.'s work mentioned in the discussion. Consider highlighting key contrasts which may offer deeper insight into potential reasons behind observed discrepancies.
Response 6: In line 315-325 we focus on the differences between ours and Bisson et al.'s work. We have elaborated on the reason behind the differences on line 323-328: Differences in AF recurrence between our study and Bisson et al. [11] may stem from treatment differences, follow-up duration and patient selection. Patients in their study underwent catheter ablation, which is a less invasive approach associated with higher recurrence rates and a greater proportion of first-time ablations, while patients in our study underwent thoracoscopic ablation. Only advanced AF patients are eligible for this procedure, resulting in less patient variability in our cohort. Additionally, our follow-up period was two years, compared to one year in the study of Bisson et al. [11].
Comment 7: Authors should mention possible future directions involving longitudinal studies following patients post-thoracoscopic ablation longer than two years. Such investigations could reveal whether initial clustering persists over extended periods.
Response 7: The reviewer makes a great point. Unfortunately, little information is available on the long term effects of ablation, having said that, we will publish results on the AFACT 10 year outcome shortly, and may consider testing this study in that cohort. We have added the following to our discussion (line 353-355): Fourth, it remains unclear whether the significance of these clusters persists over the long term. We await results from ongoing long-term follow-up studies to validate these findings.
Reviewer 2 Report
Comments and Suggestions for Authors How can identifying these clusters be implemented in daily practice? Is it going to affect patients' outcomes? If knowing patients' phenotype can be a benefit before initiating their management? Were there limitations of this study?Author Response
We thank the reviewer for their time and questions. We will answer the reviewer in a point-by-point fashion, but have grouped some questions together because of their relation to each other. We have made changes to the manuscript accordingly.
Comment 1-3: How can identifying these clusters be implemented in daily practice? Is it going to affect patients' outcomes? If knowing patients' phenotype can be a benefit before initiating their management?
Response 1-3: Despite established knowledge on AF risk factors, a considerable proportion of patients still fail AF ablation. This study aimed to identify clusters based on multiple baseline characteristics, and assess the difference in outcome or risk profile in these clusters. We found that established risk factors like overweight, AF type, and LAVI vary in importance across clusters. Knowing a patient’s cluster could discern which risk factors are most important for the individual patient and guide decisions on whether AF ablation is appropriate. The clustering model could be implemented by inputting a patient's variables into this studies PCA and K-Means model to classify them into a specific cluster.
We have added additional clarification in multiple sections:
- We found that established risk factors like BMI, AF type, and LAVI vary in importance across clusters (abstract, line 39-40).
- Moreover, we aim to assess the difference in established AF recurrence risk factors within clusters and to discern which risk factors are most relevant to each patient, thereby guiding treatment decisions (introduction, line 91-94).
- In our study, classifying a patient by its cluster, could highlight which risk factors of AF recurrence are most important for that patient (discussion, line 343-344)
Comment 4: Were there limitations of this study?
Response 4: The limitations can be found in line 344-379 of the manuscript.
Reviewer 3 Report
Comments and Suggestions for Authors
This study presents an unsupervised clustering of patients after ablation for atrial fibrillation. The described method is a machine learning technique that aims to provide insights into AF recurrence by identifying clinical features. A total of 658 patients who underwent thoracoscopic ablation between 2008 and 2021 were included in the study. In addition to the clusters, patients with and without previously identified risk factors were also compared. The unsupervised clustering resulted in three different clusters with different recurrence rates and CHA2DS2-VASc scores.
Limitations arise from the fact that initial patients and variables had to be discarded due to missing data and that the gender distribution was unbalanced. Both effects are adequately described in the discussion.
As a tip for the authors, the reviewer recommends describing in the discussion what a follow-up study might look like and what further findings could be expected from it.
Figure 2 and Figure 3: Please use different colors for the cluster (Cluster 3, Cluster 2, Cluster 1) in Figure 2 and for the factors (paroxysmal, persistent, long-lasting persistent) in Figure 3.
Author Response
We thank the reviewer for their time and advice. We will answer the reviewer in a point-by-point fashion and have made changes to the manuscript accordingly.
Comment 1: As a tip for the authors, the reviewer recommends describing in the discussion what a follow-up study might look like and what further findings could be expected from it.
Response 1: The following has been extended in the discussion (line 348-351): Further research could refine this approach by building cluster-specific treatment success prediction models, and enhancing the clustering model with a larger, more comprehensive baseline dataset, with more thorough follow-up to reduce missing data and achieve a more balanced distribution of male and female patients. Furthermore, the importance of risk factor per cluster should be tested in a validation cohort or a prospective study.
Comment 2: Figure 2 and Figure 3: Please use different colors for the cluster (Cluster 3, Cluster 2, Cluster 1) in Figure 2 and for the factors (paroxysmal, persistent, long-lasting persistent) in Figure 3.
Response 2: the colors of the factors have been changed accordingly.